# Active Components of Commonly Prescribed Medicines Affect Influenza A Virus–Host Cell Interaction: A Pilot Study

**DOI:** 10.3390/v13081537

**Published:** 2021-08-03

**Authors:** Aleksandr Ianevski, Rouan Yao, Eva Zusinaite, Hilde Lysvand, Valentyn Oksenych, Tanel Tenson, Magnar Bjørås, Denis Kainov

**Affiliations:** 1Department of Clinical and Molecular Medicine (IKOM), Norwegian University of Science and Technology, 7028 Trondheim, Norway; aleksandr.ianevski@ntnu.no (A.I.); rouan.yao@ntnu.no (R.Y.); hilde.lysvand@ntnu.no (H.L.); valentyn.oksenych@ntnu.no (V.O.); magnar.bjoras@ntnu.no (M.B.); 2Institute of Technology, University of Tartu, 50411 Tartu, Estonia; eva.zusinaite@ut.ee (E.Z.); tanel.tenson@ut.ee (T.T.); 3Institute for Molecular Medicine Finland (FIMM), University of Helsinki, 00014 Helsinki, Finland

**Keywords:** influenza virus, virus–host interaction, commonly prescribed drugs, drug adverse reaction

## Abstract

**Background:** Every year, millions of people are hospitalized and thousands die from influenza A virus (FLUAV) infection. Most cases of hospitalizations and death occur among the elderly. Many of these elderly patients are reliant on medical treatment of underlying chronic diseases, such as arthritis, diabetes, and hypertension. We hypothesized that the commonly prescribed medicines for treatment of underlying chronic diseases can affect host responses to FLUAV infection and thus contribute to the morbidity and mortality associated with influenza. Therefore, the aim of this study was to examine whether commonly prescribed medicines could affect host responses to virus infection in vitro. **Methods:** We first identified 45 active compounds from a list of commonly prescribed medicines. Then, we constructed a drug–target interaction network and identified the potential implication of these interactions for FLUAV–host cell interplay. Finally, we tested the effect of 45 drugs on the viability, transcription, and metabolism of mock- and FLUAV-infected human retinal pigment epithelial (RPE) cells. **Results:** In silico drug–target interaction analysis revealed that drugs such as atorvastatin, candesartan, and hydroxocobalamin could target and modulate FLUAV–host cell interaction. In vitro experiments showed that at non-cytotoxic concentrations, these compounds affected the transcription and metabolism of FLUAV- and mock-infected cells. **Conclusion:** Many commonly prescribed drugs were found to modulate FLUAV–host cell interactions in silico and in vitro and could therefore affect their interplay in vivo, thus contributing to the morbidity and mortality of patients with influenza virus infections.

## 1. Introduction

Thousands of drugs are currently on the market to treat diseases or improve the quality of life among those suffering from acute or chronic afflictions. Although most of these drugs confer safety and efficacy against the diseases they treat, it is impossible to avoid drugs’ unintended effects, which only appear during other diseases. These effects are often difficult to find and do not readily surface until a drug reaches post-marketing surveillance, after the drug is already approved and widely used. For example, cholesterol-lowering statins may reduce the risk of COVID-19 death [1]. By contrast, insulin use raises the risk for COVID-19 mortality in diabetes (https://www.healio.com/news/endocrinology/20210429/high-blood-glucose-at-admission-insulin-use-raise-risk-for-covid19-mortality-in-diabetes; accessed on 25 July 2021). Thus, commonly prescribed medications can affect the interaction of a virus with the host and affect morbidity and mortality.

Influenza A (FLUAV) is the most common cause of seasonal epidemics. It is responsible for 3 to 5 million cases of hospitalizations and 250,000–500,000 deaths annually. Although everyone is susceptible to FLUAV infection, most cases of hospitalizations and deaths occur among high-risk groups such as among pregnant women, young children, the elderly, and patients with immunosuppressive conditions or non-communicable diseases (NCDs) who are often reliant on various types of supportive medication when infection occurs [2]. Deaths and hospitalizations from influenza infection are often linked to underlying conditions requiring supportive or palliative medication such as asthma, diabetes, and cardiovascular and chronic kidney diseases [3].

FLUAV belongs to the family *Orthomyxoviridae*. Its genome comprises eight single-stranded viral RNA segments (vRNA) of negative polarity. Each segment interacts with three viral polymerase subunits (PA, PB1, and PB2) and nucleoproteins (NP) to make eight viral ribonucleoproteins (vRNPs). The vRNPs are surrounded by a capsid composed of matrix protein 1 (M1) and a lipid membrane derived from the host cell, embedded with hemagglutinin (HA), neuraminidase (NA), and matrix protein 2 (M2). Other viral proteins such as non-structural protein 1 (NS1), nuclear export protein (NEP), polymerase basic 1 F2 (PB1−F2), polymerase acidic X (PA−X), and the N-deleted version of PB1 (N40) are not present in the virion and only expressed in infected cells [4]. The FLUAV replication cycle consists of entry into the host cell through endocytosis, uncoating of vRNPs, import of the vRNPs into the nucleus, transcription and replication of the viral genome, translation of viral proteins, assembly of vRNPs in the nucleus, export of the vRNPs from the nucleus, assembly of virions at the host cell plasma membrane, and budding [4].

When FLUAV enters the cell, pathogen recognition receptors (PRRs) recognize viral RNA (vRNA) and initiate the transcription of interferon (IFN) genes. Once transcribed, IFNs trigger the expression of IFN-stimulated genes (ISGs) in an infected cell and, when secreted, in neighboring non-infected cells. ISGs encode different antiviral proteins including interleukins (ILs), C−X−C and C−C motif chemokines (CXCLs and CCLs), and other cytokines which recruit immune cells to the site of infection. ISGs also encode RNases, which degrade vRNAs in infected cells. NS1 hinders the cellular IFN response by binding with vRNA, cellular DNA, and other factors [5,6,7]. If a large amount of vRNA or its replication intermediates is accumulated in the cells, apoptosis is initiated. For this to occur, PRRs recognize vRNA and transduce signals to B cell lymphoma xL (Bcl−xL) protein [8,9]. Bcl−xL releases pro-apoptotic proteins to initiate mitochondrial outer membrane permeabilization (MoMP), which results in infected cell death. If IFN or apoptotic pathways are inhibited or altered, virus replication can be accelerated, and infection worsens [4].

FLUAV exploits multiple cellular factors and pathways to complete replication [10,11]. For example, cellular vATPase acidifies the interior of late endosomes. This activates cellular serine proteases that cleave HA and mediate fusion of viral and endosomal membranes. This triggers the release of vRNPs [12]. Cellular cyclin-dependent kinases (CDKs) are required for vRNA transcription, and free cellular nucleotides are used by viral polymerase to produce vRNA [13]. FLUAV also hijacks PI3K/mTor/Akt-mediated autophagy to produce free amino acids for the synthesis of viral proteins [14]. Virus assembly and budding also depend on elements of the host lipid metabolism, including de novo synthesis of cholesterol [15]. The targeting or alteration of any of these cellular factors may shift the virus–host interaction equilibrium and result in aggravation of or protection against FLUAV infection. This ability to modulate FLUAV replication has previously been utilized in the field of antiviral drug repositioning to find existing drugs that target or neutralize the necessary factors for viral replication. For example, the anticancer drug saliphenylhalamide (SaliPhe) targets vATPase, thus inhibiting FLUAV at the entry stage; the anticancer drugs flavopiridol and gemcitabine target CDKs and nucleotide metabolism which inhibit viral RNA transcription and replication, respectively; and cholesterol-lowering statins inhibit FLUAV assembly and budding [12,13,16,17]. Conversely, it has also been shown that upregulation of CDKs can increase the transcriptional activity of FLUAV vRNP and thereby can enhance the rate of viral infection [18].

As the complex nature of influenza infection involves many aspects of the host biology, we hypothesized that some of the commonly prescribed medicines could affect factors that are important for FLUAV replication, thus potentially changing the course of viral infection in patients who contract influenza while taking medication. Here, we present results from our pilot study of 45 active components of the most prescribed medicines and show how these agents affect FLUAV–host cell interaction in silico and in vitro.

## 2. Materials and Methods

### 2.1. Compounds, Cells, and Viruses

To identify the most dispensed medicines in Central Norway in 2019, we searched the Norwegian Prescription Database (www.norpd.no; accessed on 3 May 2019), which contains drugs, Anatomical Therapeutic Chemical (ATC) codes, and daily defined dosages (DDDs). As search criteria, we included all age groups and both sexes. Appendix A lists 45 active compounds of the most dispensed medicines, their suppliers, and their catalogue numbers. To obtain 10 mM stock solutions, compounds were dissolved in dimethyl sulfoxide (DMSO, Sigma-Aldrich, Steinheim, Germany) or milli-Q water, depending on solubility. The solutions were stored at −80 °C until use. Human telomerase reverse transcriptase-immortalized retinal pigment epithelial (RPE, ATCC MBA–141) cells were grown in Dulbecco’s Modified Eagle’s F12 medium (DMEM-F12; Gibco, Paisley, Scotland) supplemented with 100 U/mL penicillin and 100 ug/mL streptomycin mixture (Pen/Strep; Lonza, Cologne, Germany), 2 mM L-glutamine, 10% heat-inactivated fetal bovine serum (FBS; Lonza, Cologne, Germany), and 0.25% sodium bicarbonate (Sigma-Aldrich, St. Louis, MO, USA). Human influenza A/WSN/33(H1N1) virus was generated using the eight-plasmid reverse genetics system, as described previously [19].

### 2.2. Cell Viability Assay

Approximately 4 × 10^4^ RPE cells were seeded per well in 96-well plates. The cells were grown for 24 h in DMEM-F12 medium containing 10% FBS, and Pen/Strep. The medium was replaced with DMEM-F12 medium supplemented with 0.2% bovine serum albumin (BSA), 1 μg/mL TPSK-trypsin, and 2 mM L-glutamine. The compounds were added to the cells at seven different concentrations in 3-fold dilutions starting from 100 μM. Saliphenylhalamide (SaliPhe) and ABT–263 were used as controls [8,9,12,16]. RPE cells were infected with FLUAV at a multiplicity of infection (moi) of 1 or mock. After 48 h of infection, the medium was removed from the cells. The viability of virus- and mock- infected cells was measured using Cell Titer Glow assay (CTG; Promega, Madison, WI, USA). The luminescence was read with a PHERAstar FS plate reader (BMG Labtech, Ortenberg, Germany).

The half-maximal cytotoxic concentrations (CC_50_) and the half-maximal effective concentrations (EC_50_) for each compound were calculated using GraphPad Prism software version 7.0a. CC_50_ values were calculated based on curves obtained on mock-infected cells after non-linear regression analysis with a variable slope. The EC_50_ values were calculated based on the analysis of viability of infected cells by fitting drug dose–response curves using four-parameter (4PL) logistic function *f*(*x*):(1)$$f\left(x\right)=A_{min}+\frac{A_{max}-A_{min}}{1+{\left(\frac{x}{m}\right)}^{\lambda }}$$
where *f*(*x*) is a response value at dose *x*, *A_min_* and *A_max_* are the upper and lower asymptotes (minimal and maximal drug effects), m is the dose that produces the half-maximal effect (EC_50_ or CC_50_), and λ is the steepness (slope) of the curve. The relative effectiveness of the drug was defined as the selectivity index (SI = CC_50_/EC_50_).

### 2.3. Transcriptomics Analysis

We infected RPE cells with FLUAV at moi 1. After 8 h, we isolated total RNA using an RNeasy Plus minikit (Qiagen). A total of 384 TruSeq Stranded mRNA libraries were prepared in 96 sample batches. Sequencing was conducted on a HiSeq (HSQ-700358) instrument (set up: SR 1 × 70 bp + dual index 8 bp) using HiSeq Rapid SR Cluster Kit v2 sequencing kit, RapidRunV2 flow cell (up to 300M reads per flow cell), RTA version: 1.18.64. For the viral genome, reads were aligned to the reference influenza A/WSN/1933 using the Bowtie 2 software package version 2.3.4.1. Sequence alignments were converted to binary alignments using SAMtools version 1.5. The number of mapped and unmapped reads that aligned to each viral gene was retrieved with SAMtools idxstats. For the human genome, reads were aligned to the reference human GRCh38 genome using the Bowtie 2 software package version 2.3.4.1. The number of mapped and unmapped reads that aligned to each host gene was obtained with the featureCounts function from Rsubread R-package version 2.10.

### 2.4. Metabolomics Analysis

We infected RPE cells with FLUAV at moi 1. After 24 h, we collected the cell culture medium. We analyzed polar metabolites as described previously [20].

### 2.5. Bioinformatics Analysis

Cellular targets of drug–protein interaction were visualized using the STITCH web-tool [21]. Predicted interaction sources were excluded from the network visualization, while the line thickness was set to indicate the strength of data support.

Transcriptomics and metabolomics data were log_2_ transformed for linear modeling. The heatmaps were generated using the pheatmap package (https://cran.r-project.org/web/packages/pheatmap/index.html; accessed on 15 July 2021) based on log_2_-transformed profiling data. Gene (GSEA) and metabolite (MSEA) set enrichment analysis tools were used to retrieve pathways (http://software.broadinstitute.org/gsea/index.jsp; accessed on 15 July 2021; https://www.metaboanalyst.ca/; accessed on 15 July 2021). Structural similarity between compounds was calculated using ECPF4 fingerprints and the Tanimoto coefficient.

## 3. Results

### 3.1. Structural Comparison of 45 Active Components of Commonly Prescribed Drugs

We selected the 45 most dispensed medicines from the Norwegian Prescription Database. Appendix A lists the ATC codes, DDDs, and indications retrieved from the common catalog of pharmaceutical preparations marketed in Norway (www.felleskatalogen.no; accessed on 3 May 2019). Over-the-counter medicines, such as the anti-inflammatory ibuprofen, paracetamol, and medicines to treat allergies, heartburn, constipation, and diarrhea, were not included in our study because they are used to treat acute conditions.

Figure 1 depicts the structural relationship between 45 active components of the commonly prescribed drugs. Drugs that appear closer together on the dendrogram are more structurally related than those appearing more distant from each other. These structural relationships can often also reveal functional relationships. For example, salmeterol and salbutamol are almost structurally identical except for the absence of a long ether side chain in salbutamol. Both salmeterol and salbutamol are β2-adrenoceptor agonists which are used as smooth muscle relaxants against asthma. Other examples of structurally and functionally similar molecules include angiotensin II receptor antagonists candesartan and losartan, which are used to treat hypertension; corticosteroids mometasone furoate and fluticasone propionate, which are commonly prescribed against allergy; and proton pump inhibitors esomeprazole and pantoprazole, which are used to treat reflux and ulcers.

### 3.2. Cellular Targets of Active Components of The Most Prescribed Drugs and Their Potential Effect on FLUAV–Host Cell Interaction

We constructed an interaction network of the 45 active components connected to direct and downstream targets. The network is displayed in Figure 2, with the width and weight of the edges indicating the degree of data to support each connection. Within the network, seven targets have previously been shown to be associated with FLUAV replication. These were CXC chemokine receptor type 4 (CXCR4), albumin precursor (ALB), histamine receptor H1 (HRH1), Ras homolog family member A (RHOA), alpha-2B adrenergic receptor (ADRA2B), purine nucleoside phosphorylase (PNP), and metabolism of cobalamin associated B (MMAB) proteins. These factors are targeted by niacin, losartan, ramipril, acetylsalicylic acid, thyroxine, valsartan, cetirizine, citalopram, atorvastatin, simvastatin, metoprolol, sertraline, candesartan, and hydroxocobalamin [10,11]. Thus, commonly prescribed drugs could modulate FLUAV–host cell interactions and therefore could affect the morbidity and mortality of influenza-infected patients.

### 3.3. Toxicity and Efficacy of Active Components of Commonly Prescribed Medicines in Mock- and FLUAV-Infected RPE Cells

In order to examine the effect of the active components of commonly prescribed medicines on the viability of RPE cells, the cells were treated with compounds in 3-fold dilutions at seven different concentrations starting from 100 μM. No compounds were added to the control wells. After 48 h, cell viability was measured. The half-maximal cytotoxic concentrations of each compound were determined and plotted (Figure 3). We found that all compounds, except eight, were nontoxic at the tested range of concentrations (<100 μM). The eight compounds that demonstrated toxicity were amlodipine (CC_50_ = 28.5 μM), desloratadine (CC_50_ = 46.7 μM), desogestrel (CC_50_ = 34.5 μM), salmeterol (CC_50_ = 29.3 μM), sertraline (CC_50_ = 17.5 μM), simvastatin (CC_50_ = 48.3 μM), vitamin D2 (CC_50_ = 42.1 μM), and vitamin D3 (CC_50_ = 48.8 μM).

We also determined the antiviral efficacy of the active components. For this, RPE cells were treated with compounds in 3-fold dilutions at seven different concentrations starting from 100 μM and infected with FLUAV at an moi of 1. No compounds were added to the control wells. After 48 h, cell viability was measured. The half-maximal effective concentrations of each compound were determined. We found that none of the compounds rescued cells from FLUAV-mediated death.

### 3.4. Active Components of Commonly Prescribed Medicines Affect Gene Expression in Mock- and FLUAV-Infected RPE Cells

We evaluated the effects of the compounds on the transcription of host genes in mock- and virus-infected RPE cells. The cells were either treated with 10 μM of drug or remained untreated, and either infected with mock or FLUAV (moi 1). After 8 h, we isolated total RNA and sequenced the polyadenylated fraction of RNA. We constructed a heatmap of the most variable genes of mock- and FLUAV-infected cells (Figure 4 and Figure 5). Almost all compounds affected the transcription of the host genes in both uninfected and FLUAV-infected RPE cells. Interestingly, structurally similar drugs salmeterol and salbutamol both increased the expression of phosphodiesterase 4D (PDE4D), cysteine-rich secretory protein LCCL domain-containing 2 (CRISPLD2), and prostaglandin E synthase (PTGES) and decreased the expression of solute carrier family 26 member 4 (SLC26A4), family with sequence similarity 111, member B (FAM111B), and cyclin E2 (CCNE2) in non-infected cells compared to the levels found in non-treated cells, but they did not affect FLUAV-activated host gene expression. This suggests that drugs administered during FLUAV infection may affect gene expression abnormally, thus changing the overall activity of the genes Notably, several compounds including amlodipine, drospirenone, esomeprazole, lercanidipine, sertraline, and simvastatin also increased the expression of prostaglandin-endoperoxide synthase 2 (PTGS2), a gene known to play a role in the resolution of both infectious and non-infectious inflammation, in both mock- and virus-infected cells.

We also analyzed viral polyadenylated RNAs in drug-treated and non-treated cells (Figure 6). Atorvastatin, cetirizine, acetylsalicylic acid, nicotinic acid, naproxen, tamsulosin, thiamine, and vitamin D3 attenuated the transcription of all viral RNAs (fold change >2 log_2_). Interestingly, atorvastatin and cetirizine share structural similarities, as do acetylsalicylic acid and nicotinic acid, indicating that there is a structure–activity relationship of drugs that decreases viral gene expression. By contrast, metformin differentially affected polyadenylated viral RNAs, increasing the expression of some while decreasing the expression of others. Taken together, we can see that the use of these medications can indeed affect both host and viral gene expression.

### 3.5. Active Components of Commonly Prescribed Medicines Affect Metabolism of Mock- and FLUAV-Infected RPE Cells

Next, we evaluated the effect of the compounds on the metabolism of mock- and virus-infected RPE cells. The cells were either treated with 10 μM of drug or remained untreated, and either mock- or FLUAV-infected (moi 1). After 24 h, the supernatants were collected, 102 polar metabolites were analyzed. We constructed a heatmap of the most variable metabolites (Figure 7 and Figure 8). Of note, we found that amlodipine substantially elevated the levels of trimethylamine-*N*-oxide, adenine, NAD, cytosine, octanoylcarnitine, and homocysteine in the media of mock-infected cells, as well as inosine and D-ribose 5-phosphate in the media of FLUAV-infected cells. We also found that levonorgestrel substantially lowered the levels of trimethylamine-*N*-oxide, phosphoethanolamine, and hippuric acid in non-infected cells, but not in FLUAV-infected cells.

Other drugs deregulated the metabolism of different polar molecules in the media of mock- and FLUAV-infected cells, but to a lesser extent. For example, 17a-ethynylestradiol, 4-acetamidophenol, acetylsalicylic acid, amlodipine, atorvastatin, bumetanide, candesartan, and cetirizine elevated the levels of many polar metabolites in the media of infected cells.

FLUAV infection itself was found to increase the concentration of L-kynurenine and inosine and lower the concentration of spermidine and NAD in agreement with previous studies [20,22,23], but treatments only moderately affected the levels of these metabolites. From this, we can see that many commonly prescribed medicines can, indeed, affect the metabolic activity of both mock- and FLUAV-infected cells.

## 4. Discussion

The pharmaceutical industry is one of the most strictly regulated industries in the world, ensuring that medicines approved with marketing authorization are safe and effective, and that the benefits of the drugs outweigh the potential risks to the patients. Here, we investigated whether commonly prescribed medicines could affect FLUAV–host cell interactions.

Of the 45 active compounds in commonly prescribed medicines in Norway, we found that niacin, losartan, ramipril, acetylsalicylic acid, thyroxine, valsartan, cetirizine, citalopram, atorvastatin, simvastatin, metoprolol, sertraline, candesartan, and hydroxocobalamin may target human genes known to be involved in FLUAV infection. Furthermore, we found that amlodipine, drospirenone, esomeprazole, lercanidipine, sertraline, simvastatin, acetylsalicylic acid, atorvastatin, candesartan, and hydroxocobalamin strongly affected the transcription and metabolism of both mock- and FLUAV-infected cells. Specifically, amlodipine, drospirenone, esomeprazole, lercanidipine, sertraline, and simvastatin were all shown to increase the expression of PTGS2, a gene known to play a role in the resolution of infectious and non-infectious inflammation and therefore likely to play a role in FLUAV infection. Taken together, this work shows that many common prescription drugs can modulate FLUAV–host cell interaction. More work needs to be conducted to determine if this modulation is to an extent that could impact the progression and severity of disease.

Interestingly, we found that the structurally analogous drugs salmeterol and salbutamol were able to strongly modulate the expression of several host genes in mock-infected cells but were unable to achieve the same action in FLUAV-infected cells. This was further echoed in our metabolomics analysis, which showed that levonorgestrel substantially altered the levels of certain metabolites in mock-infected cells but did not cause the same change in metabolite levels for FLUAV-infected cells. Similarly, we found that the substantial elevation that amlodipine brought to certain polar metabolite levels in mock-infected cells was replaced by a substantial elevation in other polar metabolites when treating FLUAV-infected cells. This suggests that not only can common drugs impact the progression and severity of FLUAV infection but also the pharmacodynamic actions of the drugs themselves may be impacted by the presence of a concurrent infection. Due to widespread use of the common medicines in our study and the pervasiveness of influenza each year, we hope that this potential change in the pharmacological action of drugs may be taken into consideration and studied further.

It is important to note that the current study is a proof-of-concept pilot study that reports on results using only one line of human nonmalignant RPE cells and one laboratory H1N1 strain. While these tools provided a good starting point, we hope to expand this investigation to cover more applicable models in the context of FLUAV infection, such as in human macrophages and dendritic cells, or in animal models. Additionally, we are interested in whether our observations will hold when testing against shifting seasonal FLUAV strains and other viruses, such as SARS-CoV-2. Finally, because our study of transcriptomics and metabolomics yielded interesting results, we hope to expand our experimental repertoire to investigate these drugs’ impacts on the virus–host networks in the context of epigenetics, proteomics, phosphoproteomics, and lipidomics. This can not only give a fuller picture of the true impact of viral infection and medicine use on each other but can also allow for the identification of -omics signatures that are associated with significant medicine–infection interactions to allow for easier discovery of drug side effects associated with virus infections.

## 5. Conclusions

There are thousands of approved drugs currently being used around the world, and more experimental and investigational drugs are being developed daily. Many of these drugs have side effects which can only be revealed in clinical studies or long-term post-market monitoring and can be challenging and time-consuming to identify. Here, we used in silico and in vitro approaches [24] to study the effects of 45 commonly prescribed medicines on FLUAV–host cell interaction. By simultaneously leveraging drug–target interaction studies, drug toxicity/efficacy tests, transcriptomics, and metabolomics, we were able to observe drug-induced transcriptional and metabolomic changes in the context of both FLUAV and mock infection. Thus, we outline a straightforward, in silico–in vitro method to identify hidden cross-over effects of common medications. Overall, we believe that our systems biology approach could be applied broadly in the pharmaceutical industry during drug development.

## Figures and Tables

**Figure 1 viruses-13-01537-f001:**
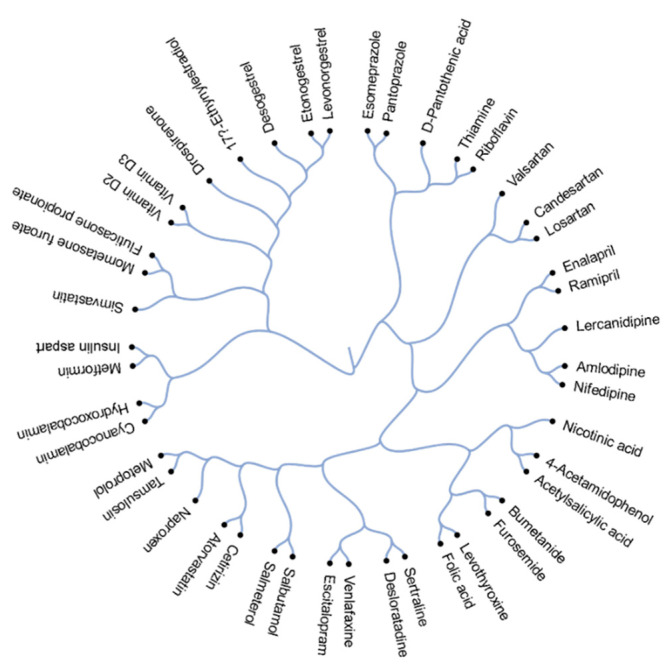
Active compounds of 45 most prescribed drugs clustered based on their structural similarity, calculated by ECPF4 fingerprints and the Tanimoto coefficient.

**Figure 2 viruses-13-01537-f002:**
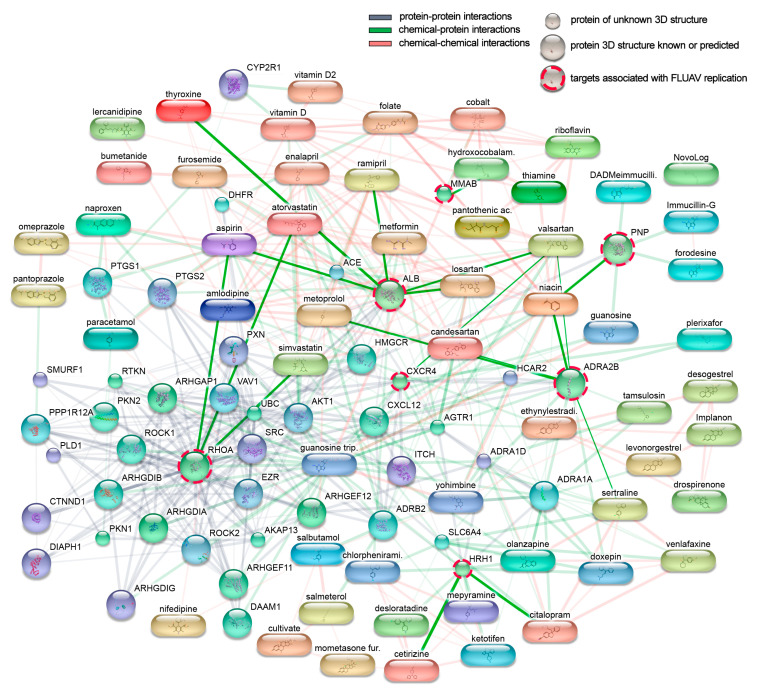
Direct and downstream cellular targets of 45 active components of commonly prescribed medicines. Targets associated with FLUAV replication are marked with red dashed circles, and interactions between them and commonly prescribed drugs are highlighted.

**Figure 3 viruses-13-01537-f003:**
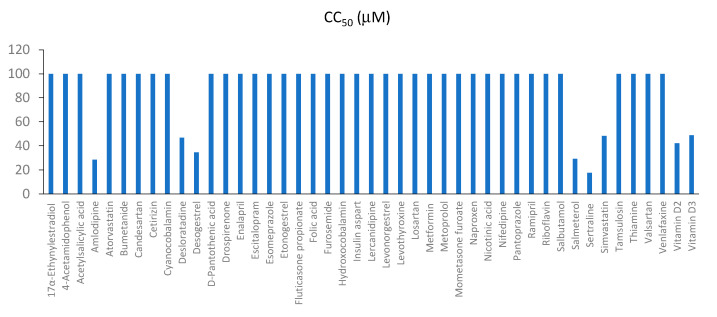
Effect of 45 active compounds of commonly prescribed drugs on the viability of mock- and FLUAV-infected RPE cells. CC_50_ and EC_50_ values are shown in blue and orange, respectively (mean, *n* = 3). CC_50_ = 100 means >100 μM.

**Figure 4 viruses-13-01537-f004:**
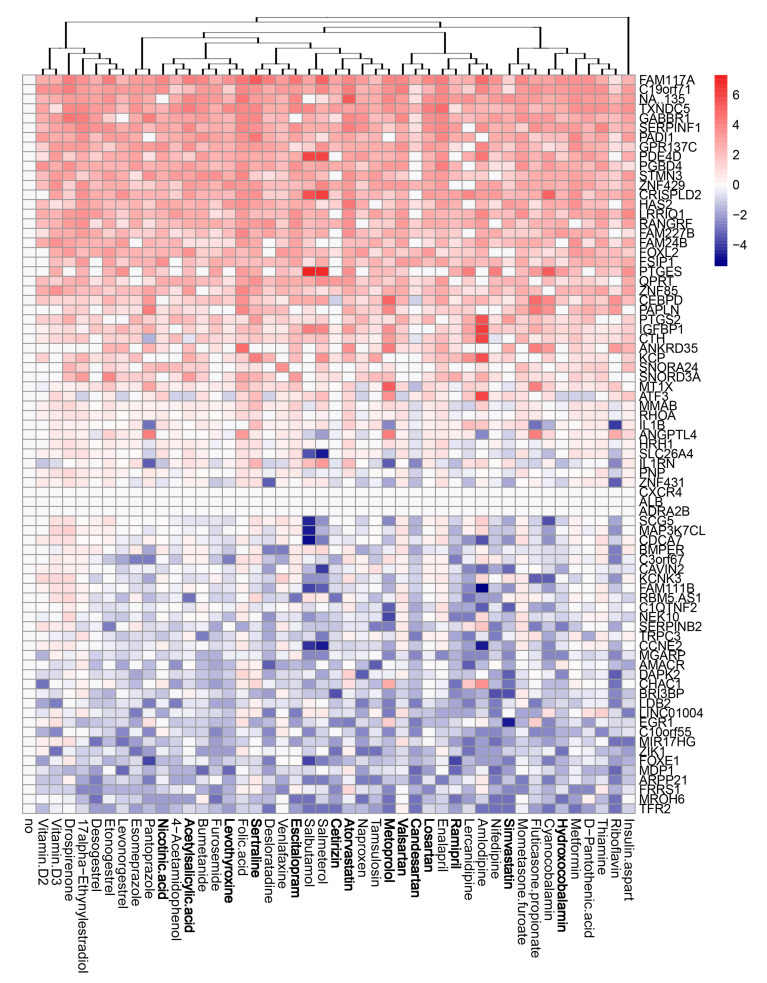
Effect of 45 compounds on polyadenylated host RNAs in RPE cells. RPE cells were treated with 10 µM compounds or remained non-treated. After 8 h, total RNA was isolated, and a fraction of polyadenylated RNA was sequenced. A heatmap of 70 most variable mRNAs affected by treatment is shown. Rows represent gene symbols, columns represent treatments with drugs, clustered based on structural similarity. Drugs marked in bold are compounds which were predicted to affect FLUAV–host cell interactions. Each cell is colored according to the log2-transformed expression values of the samples, expressed as fold change relative to the non-treated control.

**Figure 5 viruses-13-01537-f005:**
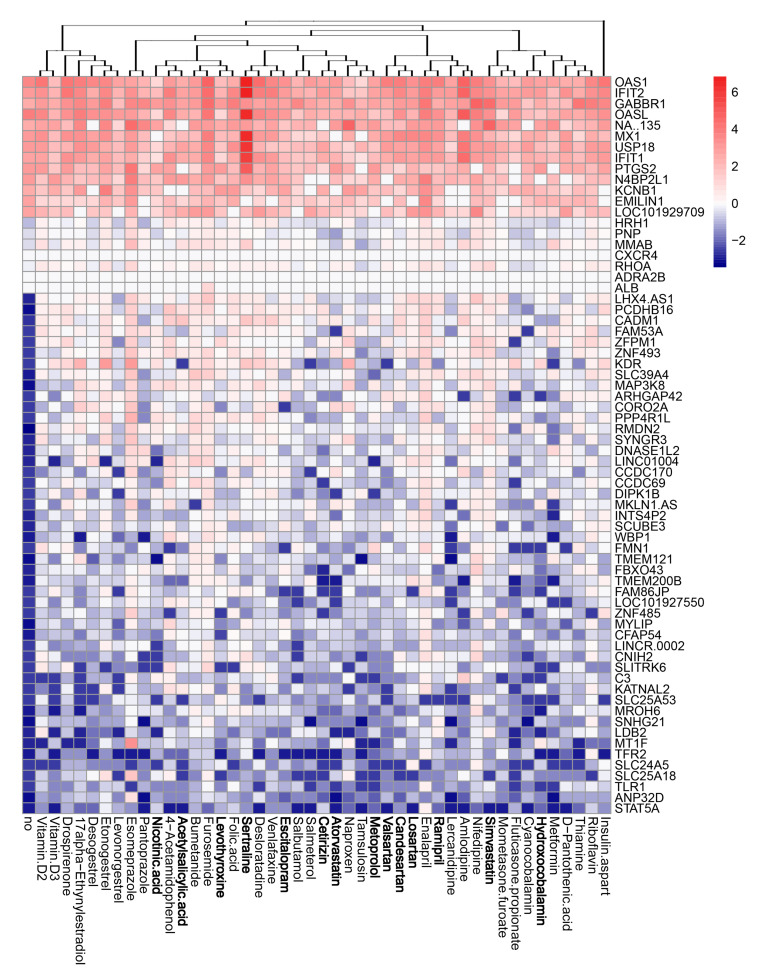
Effect of 45 compounds on polyadenylated host RNAs in FLUAV-infected RPE cells. RPE cells were treated with 10 µM compounds and infected with FLUAV at moi 1. After 8 h, total RNA was isolated, and a fraction of polyadenylated RNA was sequenced. A heatmap of the most variable genes affected by FLUAV infection is shown (2.5 < log2FC < −2.5). Rows represent gene symbols, columns represent treatments with drugs, clustered based on structural similarity. Drugs marked in bold are compounds which were predicted to affect FLUAV–host cell interactions. Each cell is colored according to the log2-transformed expression values of the samples, expressed as fold change relative to the non-treated mock-infected control.

**Figure 6 viruses-13-01537-f006:**
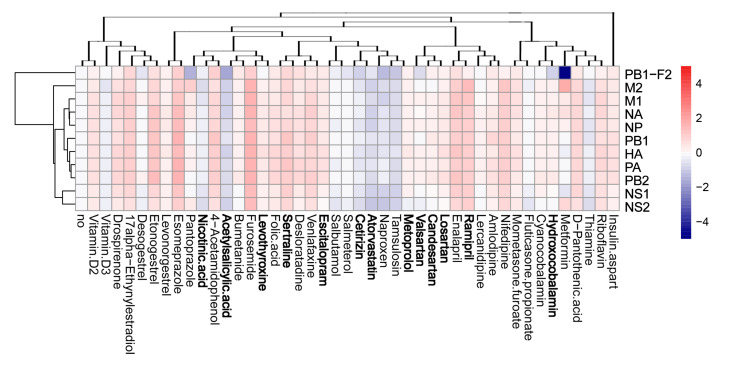
Effect of 45 compounds on polyadenylated viral RNAs in FLUAV-infected RPE cells. RPE cells were treated with 10 µM compounds or remained non-treated and infected with FLUAV at moi 1. After 8 h, total RNA was isolated, and a fraction of polyadenylated RNA was sequenced. A heatmap of viral RNAs affected by treatment is shown. Rows represent gene symbols, columns represent treatments with drugs, clustered based on structural similarity. Drugs marked in bold are compounds which were predicted to affect FLUAV–host cell interactions. Each cell is colored according to the log2-transformed expression values of the samples, expressed as fold change relative to the non-treated FLUAV-infected control.

**Figure 7 viruses-13-01537-f007:**
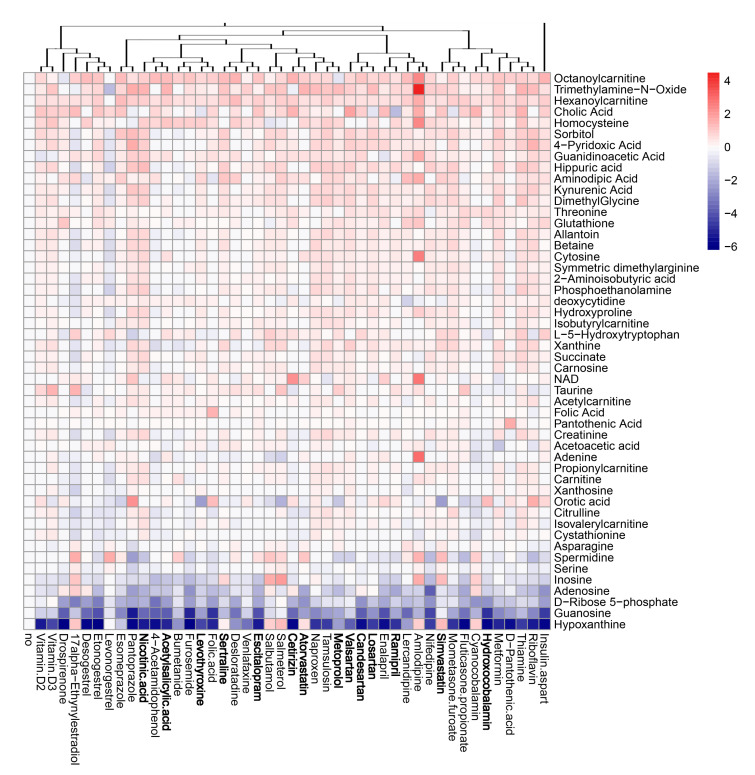
Effect of 45 compounds on metabolism of polar molecules in mock-infected RPE cells. The cells were treated with 10 µM compounds or remained non-treated. After 24 h, the media were collected, and polar metabolites were analyzed using LC-MS/MS. A heatmap of 50 most variable metabolites affected by treatment is shown. Rows represent metabolites, columns represent treatments with drugs, clustered based on structural similarity. Drugs marked in bold are compounds which were predicted to affect FLUAV–host cell interactions. Each cell is colored according to the log2-transformed and quantile-normalized values of the samples, expressed as fold change relative to the non-treated control.

**Figure 8 viruses-13-01537-f008:**
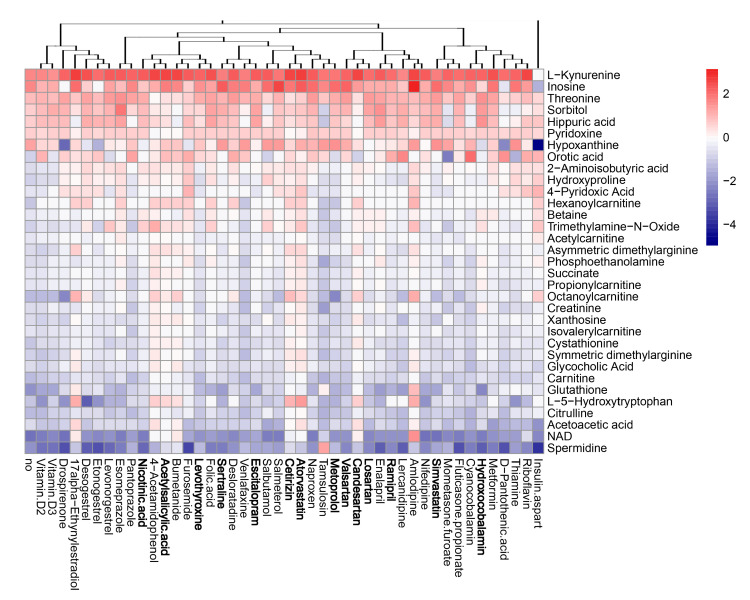
Effect of 45 compounds on metabolism of FLUAV-infected RPE cells. The cells were treated with 10 µM compounds or remained non-treated. After 24 h, the media were collected, and polar metabolites were analyzed using LC-MS/MS. A heatmap of 33 most variable metabolites affected by treatment is shown. Rows represent metabolites, columns represent treatments with drugs, clustered based on structural similarity. Drugs marked in bold are compounds which were predicted to affect FLUAV–host cell interactions. Each cell is colored according to the log2-transformed and quantile-normalized values of the samples, expressed as fold change relative to the non-treated mock-infected control.

## Data Availability

Data supporting the reported results can be requested from the corresponding author.

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
