# Peer review of "Active Components of Commonly Prescribed Medicines Affect Influenza A Virus–Host Cell Interaction: A Pilot Study"

_viruses, 2021, doi:10.3390/v13081537_

Round 1

Reviewer 1 Report

Based on the hypothesis that some treatments of underlying chronic diseases could affect pathogenesis of Influenzavirus, the authors have investigated host responses to commonly prescribed medicines. Specifically, they explore the effect of 45 active components of the most prescribed medicines on virus-host interaction in silico and in vitro. Interestingly, none of the drugs tested seem to have a direct restrictive effect on influenza virus replication (according to EC50 illustrated in Fig3).

This kind of investigations are necessary to better define treatments to viral infections, since underlying conditions such as a disease but also the corresponding treatment can alter infection progression. It becomes essential to find adequate and specific treatments from a holistic view of the patient.

General points

  • The study has been only performed with one influenza virus strain (A/WSN/33(H1N1) virus). It would be interesting to investigate if these drugs could have similar effect on replication of other influenza virus strains, including Influenza A from last pandemics and from Influenza B virus (that is normally circulating and responsible for many of Flu infections).
  • It is difficult to draw conclusions from the Heatmaps included in figures 4, 5, 6, 7 and 8. The authors could should the effect of most relevant drugs individually on all aspects investigated in this study. Also, toxic drugs (according to Fig 3) should not be included in further analysis.
  • The authors could show more clearly if drugs structurally similar (according to fig1) show similar effects on functional assays.
  • The authors could represent more clearly if drugs identified as interactors to cellular proteins required for influenza virus replication (highlighted in Fig 2) show more or less effect on virus replication than the rest of the drugs included in the study.
  • The authors could expand the discussion section to better explain the impact of their investigation and how they would perform this kind of investigation in clinical studies.
  • In the conclusions sections it is said “we were able to identify individual markers associated with potential adverse drug reactions related to influenza virus infection”. However, there are no clear statements of which are these markers and how certain drugs alter influenza infection progression in the patients.

Author Response

R.1.1. The study has been only performed with one influenza virus strain (A/WSN/33(H1N1) virus). It would be interesting to investigate if these drugs could have similar effect on replication of other influenza virus strains, including Influenza A from last pandemics and from Influenza B virus (that is normally circulating and responsible for many of Flu infections).

Re: As mentioned by second reviewer, our study is exploratory concerning possible unintended effects from the usage of common medications for non-influenza-related diseases during influenza infection.  We utilized a combination of bioinformatics, cell viability, transcriptomics and metabolomics assays to determine potential host and viral targets of these medications.  The is a pilot study which predicts and confirms effects of medications on A/WSN/33(H1N1)-RPE cell interaction.  We now discussed these points in more detail: ‘It is important to note that the current study is a proof-of-concept pilot study that reports on results using only one line of human nonmalignant RPE cells and one laboratory H1N1 strain. While these tools provided a good starting point, we hope to expand this investigation to cover more applicable models in the context of FLUAV infection, such as in human macrophages and dendritic cells, or in animal models. Additionally, we are interested in whether our observations will hold when testing against shifting seasonal FLUAV strains and other viruses, such as SARS-CoV-2. Finally, because our study of transcriptomics and metabolomics yielded interesting results, we hope to expand our experimental repertoire to investigate these drugs’ impacts on the virus-host networks in the context of epigenetics, proteomics, phosphoproteomics and lipidomics. This can not only give a fuller picture of the true impact of viral infection and medicine use on each other, but also allow for the identification of -omics signatures that are associated with significant medicine-infection interactions to allow for easier discovery of drug side effects associated with virus infections.’

R.1.2. It is difficult to draw conclusions from the Heatmaps included in figures 4, 5, 6, 7 and 8. The authors could show the effect of most relevant drugs individually on all aspects investigated in this study. Also, toxic drugs (according to Fig 3) should not be included in further analysis.

Re: We used non-toxic drug concentrations (10 μM) in transcriptomics and metabolomics assays. We have modified the heatmaps shown in Fig. 4-8, in accordance with the structural similarity of the drugs.

R.1.3. The authors could show more clearly if drugs structurally similar (according to fig1) show similar effects on functional assays.

Re: We have now grouped the drugs based on structural similarity (according to Fig. 1) to more clearly show their effect on mock- and FLUAV-infected RPE cells in functional analyzes.

R.1.4. The authors could represent more clearly if drugs identified as interactors to cellular proteins required for influenza virus replication (highlighted in Fig 2) show more or less effect on virus replication than the rest of the drugs included in the study.

Re: We have now highlighted in bold the drugs identified in silico as inhibitors of FLUAV-host cell interactions in Fig. 4-8.

R1. 5. The authors could expand the discussion section to better explain the impact of their investigation and how they would perform this kind of investigation in clinical studies.

Re: The discussion section has been expanded to discuss the impact of our findings and spell our future directions.  

R.1.6. In the conclusions sections it is said “we were able to identify individual markers associated with potential adverse drug reactions related to influenza virus infection”. However, there are no clear statements of which markers these are and how certain drugs alter influenza infection progression in the patients.

Re: The sentence has been changed to clarify our meaning – that we observed drug-induced alteration of transcription/metabolism in the context of viral infection, but cannot comment in how they will affect infection in patients.

Reviewer 2 Report

This manuscript from Ianevski et al describes exploratory studies concerning possible unintended effects from the usage of common medications for non-influenza-related diseases during influenza infection.  This group utilized a combination of bioinformatics, cell viability and toxicity assays and viral and gene host expression to determine potential host and viral targets of these medications.  The study is observational by nature given its status as a pilot study and unbiased reporting of predicted targets and effects on host targets.  This being taken into account, I have the following comments that should be addressed prior to acceptance for publication:

1) The abstract and introduction are somewhat misleadingly written to imply that the authors are testing these medications as possible novel drugs to be used in influenza treatment rather than to test effects when being taken during influenza infection for other conditions- I would suggest these sections to be rewritten to clarify this point. 

2) In section 3.2, the text misleadingly suggests that figure 2 demonstrates that the targets outlined in red could impact influenza replication when figure 2 (and the description for figure 2) is really demonstrating that host factors that were previously found to impact influenza replication are predicted to be targets of these drugs.  Section 3.2 should be rewritten to reflect this.

3) In section 3.3 the authors should specify how their observed LD50 and cellular toxicity values compare to previous studies.  The authors could also include the data from the cell toxicity assays to quantify their observation that these drugs didn't rescue cells from influenza-induced death.

4) Figure 3 is too crowded given the number of compounds and the fact that the EC50 for all of them is reported as 100 uM- this would be better represented as a table or as a bar graphs without the EC50.

5) In section 3.4 the authors should explain the significance of the factors they've highlighted in 233-237.

6) None of the figures identified as potential targets in figure 2 were included in the RNA sequencing heatmaps in figures 4 and 5- this would be a nice inclusion to see how closely the bioinformatics study predicted behavior in these cells.

7)  The heatmap in figure 6 has a misleading color bar since it has equal boldness to 1X increases and 5X decreases, making the increases in viral RNA seem stronger than they actually are.  Also in lines 259-268, the authors should specify whether the drugs that increased viral RNA levels did so to any degree of significance, as this is one potential side effect of these drugs that this paper aims to find.

8) The authors should explain the significance of the metabolites highlighted in section 3.5, and should also make it clearer whether they are referring to the mock-infected or FLUAV infected cells.

Author Response

R2.1. The abstract and introduction are somewhat misleadingly written to imply that the authors are testing these medications as possible novel drugs to be used in influenza treatment rather than to test effects when being taken during influenza infection for other conditions- I would suggest these sections to be rewritten to clarify this point. 

Re: The language in the abstract has been tweaked to reflect the findings of the paper more clearly. The introduction has been reorganized and changed to better introduce and reflect the overall goal of the paper.

R2.2. In section 3.2, the text misleadingly suggests that figure 2 demonstrates that the targets outlined in red could impact influenza replication when figure 2 (and the description for figure 2) is really demonstrating that host factors that were previously found to impact influenza replication are predicted to be targets of these drugs.  Section 3.2 should be rewritten to reflect this.

Re: This has now been changed to clearly state that factors mentioned have been previously found to impact influenza replication.

R.2.3. In section 3.3 the authors should specify how their observed LD50 and cellular toxicity values compare to previous studies. The authors could also include the data from the cell toxicity assays to quantify their observation that these drugs didn't rescue cells from influenza-induced death.

Re: We searched PubMed but could not find EC50 values for the 45 drugs used in our study. We've also simplified fig. 3, but mentioned in the text that none of the drugs saved RPE cells from death mediated by FLUAV.

R.2.4. Figure 3 is too crowded given the number of compounds and the fact that the EC50 for all of them is reported as 100 uM- this would be better represented as a table or as a bar graphs without the EC50.

Re: We agree. We have now excluded EC50 values from the graph (Fig.3).

R.2.5. In section 3.4 the authors should explain the significance of the factors they've highlighted in 233-237.

Re: By listing the genes in 233-237, we were simply trying to highlight the curious fact that several genes were shown to be strongly upregulated or downregulated in uninfected cells but showed no change in FLUAV-infected cells. Because the genes themselves have wide and generalized physiological actions that are generally not considered relevant in viral replication, we feel that describing their individual significance is not relevant enough to include in this section. However, we did add a sentence explaining the significance of our observation.

R.2.6. None of the figures identified as potential targets in figure 2 were included in the RNA sequencing heatmaps in figures 4 and 5- this would be a nice inclusion to see how closely the bioinformatics study predicted behavior in these cells.

Re: We have now included potential targets identified in Fig.2 into Fig. 4 and 5.

R.2.7.  The heatmap in figure 6 has a misleading color bar since it has equal boldness to 1X increases and 5X decreases, making the increases in viral RNA seem stronger than they actually are.  Also in lines 259-268, the authors should specify whether the drugs that increased viral RNA levels did so to any degree of significance, as this is one potential side effect of these drugs that this paper aims to find.

Re: We have now adjusted the color bar in Fig. 6 (-5<log2<+5). There are few drugs that slightly increased viral RNA levels but it’s impossible to estimate the significance and logFC difference for them is rather low (<1.5).

R.2.8. The authors should explain the significance of the metabolites highlighted in section 3.5, and should also make it clearer whether they are referring to the mock-infected or FLUAV infected cells.

Re: Our analysis was limited to 102 polar metabolites. We identified 50 and 33 most variable metabolites in mock- and FLUAV-infected RPE cells, respectively. These metabolites belong to different metabolic pathways, including tryptophane, purine, glutathione, nitrogen, arginine and proline, alanine, asparagine and glutamine, histamine, cysteine and methionine pathways. We have now made this clear in the text of section 3.5.